# Grape SnRK2.7 Positively Regulates Drought Tolerance in Transgenic *Arabidopsis*

**DOI:** 10.3390/ijms25084473

**Published:** 2024-04-18

**Authors:** Guanquecailang Lan, Weifeng Ma, Guojie Nai, Guoping Liang, Shixiong Lu, Zonghuan Ma, Juan Mao, Baihong Chen

**Affiliations:** College of Horticulture, Gansu Agricultural University, Lanzhou 730070, China

**Keywords:** grapes, drought stress, glucose metabolism, yeast two-hybrid assay, functional verification

## Abstract

In this study, we obtained and cloned *VvSnRK2.7* by screening transcriptomic data to investigate the function of the grape sucrose non-fermenting kinase 2 (SnRK2) gene under stress conditions. A yeast two-hybrid (Y2H) assay was used to further screen for interaction proteins of VvSnRK2.7. Ultimately, VvSnRK2.7 was heterologously expressed in *Arabidopsis thaliana*, and the relative conductivity, MDA content, antioxidant enzyme activity, and sugar content of the transgenic plants were determined under drought treatment. In addition, the expression levels of *VvSnRK2.7* in *Arabidopsis* were analyzed. The results showed that the VvSnRK2.7-EGFP fusion protein was mainly located in the cell membrane and nucleus of tobacco leaves. In addition, the VvSnRK2.7 protein had an interactive relationship with the VvbZIP protein during the Y2H assay. The expression levels of *VvSnRK2.7* and the antioxidant enzyme activities and sugar contents of the transgenic lines were higher than those of the wild type under drought treatment. Moreover, the relative conductivity and MDA content were lower than those of the wild type. The results indicate that *VvSnRK2.7* may activate the enzyme activity of the antioxidant enzyme system, maintain normal cellular physiological metabolism, stabilize the berry sugar metabolism pathway under drought stress, and promote sugar accumulation to improve plant resistance.

## 1. Introduction

Fruit trees are subjected to abiotic stresses including salinity, low temperature, and drought during their growth cycles, which seriously affects their growth and development, as well as fruit quality and yield [1,2]. However, over a long period of evolution, fruit trees can develop a variety of stress response mechanisms, such as regulating the cellular production of favorable substances or activating the expression of relevant genes to resist a variety of abiotic stresses, thereby maintaining normal plant growth [3,4,5].

Among these mechanisms, reversible protein phosphorylation, coordinated by protein phosphatases and kinases, is pivotal for sensing and responding to abiotic stresses and represents a key mechanism for controlling cellular functions, including responses to phytohormones, pathogens, and environmental stimuli, as well as metabolic regulation in plants [6]. In plants and green algae, SNF1 (sucrose non-fermenting1)-associated protein kinase (*SnRK*) is a central component of an evolutionarily conserved kinase cascade [6]. However, research on the function of the grape *SnRK2* gene in grape drought resistance remains underdeveloped. The group mined *SnRK2.7* from the transcriptome data of ‘Italian Riesling’ grapes to perform drought resistance research and explored the expression pattern of grape *SnRK2.7* and other genes. These data are useful for investigating the regulatory mechanism of *SnRK2.7* in response to drought stress.

The plant *SnRK* family can be divided into three subfamilies, *SnRK1*, *SnRK2*, and *SnRK3*. *SnRK2* and *SnRK3* are a unique class of serine/threonine protein kinases in plants [6,7], which play major roles in plant ABA-mediated signaling, response to drought stress, and plant resistance enhancement [6]. *CIPK* (CBL-interacting protein kinase) is an intercalating protein kinase belonging to the *SnRK3* family of calcium-regulated phosphatase B-like proteins [8]. In addition, drought induces phosphorylation of the MdCIPK22 protein kinase and interacts with MdSUT2.2, which promotes the accumulation of soluble sugars in the vesicles and enhances plant drought resistance [9,10]. An analysis of gene expression patterns revealed that tobacco *NtSnRK2* genes significantly respond to abiotic stresses such as salt, drought, and low-temperature stress, suggesting their extensive involvement in the adaptation of tobacco to unfavorable environmental conditions [1]. In addition, the heterologous expression of maize *SAPK8* and rice *SAPK4* in *Arabidopsis* improves salt tolerance in *Arabidopsis* [11,12]. It was shown that the ABA-activated protein kinase is phosphorylated in response to ABA signaling and regulates gene expression. This kinase plays a positive regulatory role in ABA and stress signaling. The SnRK2 protein kinase is involved in stress signaling responses in plants and plays an important role in enhancing abiotic stress tolerance through ABA-dependent and ABA-independent pathways [13]. Osmotic stress also activates ABA-independent responses through *SnRK2* [14]. Giribaldi et al. [15] found that ABA affects the activity of vesicle invertase (GIN1) during fruit ripening. In addition, ABA activates the downstream signaling component ABI4, which regulates the expression of sugar-responsive genes, suggesting that activation of the ABA-independent response by *SnRK2* can regulate the glucose metabolic response by signaling to the nucleus and inducing the expression of relevant genes [16,17,18]. In conclusion, the *SnRK2* gene responds to abiotic stress and regulates sugar metabolism processes under abiotic stress, but there have been no detailed studies on the biological functions of grape *SnRK2* genes under abiotic stress. Researchers found that *SnRK2* genes regulate drought stress tolerance in plants and drought resistance through ABA-dependent and ABA-independent pathways.

*VvSnRK2.7* was screened from the transcriptome data of Italian Riesling grapes provided in a previous study [19], but whether *VvSnRK2.7* is related to drought stress remains unclear and must be further explored. In this experiment, grape *VvSnRK2.7* was overexpressed in *Arabidopsis*. Y2H screening for reciprocal proteins and subcellular localization was performed to study the expression position of *VvSnRK2.7* to further confirm its function and provide future theoretical support for the mining and utilization of the gene.

## 2. Results Analysis

### 2.1. Evolutionary Analysis of VvSNRK2 Genes

To investigate the interspecific evolutionary relationships of grape *SnRK2.7*, we used MEGA 7.0 software (Figure 1) to construct a phylogenetic tree of the *SnRK2* gene in six species: Pinot Noir, Chardonnay, Sandy Grape, maize, strawberry, and *Arabidopsis*. We found that the gene family could be classified into five subfamilies, subfamilies A, B, C, D, and E, which contained 1, 2, 17, 12, and 20 members, respectively. Further analysis revealed that *VvSnRK2*.7 was closely related to *ChSnRK2*.8 and *VrSnRK2*.7. The results of a multiple sequence alignment (Figure 2) showed that the amino acid sequences of seven *VvSnRK2* members are highly conserved. Additionally, the amino acid sequences of the *SnRK2* gene family contain two structural N-terminal and C-terminal domains. The N-terminus is the catalytic domain, which is highly conserved and contains the ATP-binding domain and serine/threonine protein kinase active site; the N-terminus is also the core catalytic region of *SnRK2.* The C-terminus consists of abiotic stress-response domains and ABA-induced regulatory domains.

### 2.2. Analysis of Physicochemical Properties of VvSnRK2.7

We analyzed three varieties of Pinot Noir, Sandy Grape, and Chardonnay to understand differences in the physicochemical properties of the grape VvSnRK2.7 protein (Table 1). We found that Pinot Noir (*VvSnRK2.7*), Chardonnay (*ChSnRK2.2*), and Sandy Grape (*VrSnRK2.1*) had the same number of amino acids: 355. Moreover, the molecular weights were smallest for *VvSnRK2*.7 and largest for *ChSnRK2*.1, with 40.61 and 40.68 kD, respectively. In addition, the isoelectric point and instability index were within the ranges 6.05~6.1 and 47.07~47.12. In summary, *VvSnRK2*.7, *ChSnRK2.1,* and *VrSnRK2.1* have similar physicochemical properties.

### 2.3. Subcellular Localization of VvSNRK2.7

The VvSnRK2.7-pCAMBIA1300-GFP recombinant plasmid containing *VvSnRK2.7* was introduced into *Agrobacterium tumefaciens* GV3101 and transformed into tobacco for subcellular localization observation, indicating that *VvSnRK2.7* was localized in the cell membrane and nucleus (Figure 3).

### 2.4. Yeast Library Screening and PCR Assay

Ninety-eight monoclonal lines were screened from a yeast library using the pGBKT7-*VvSnRK2.7* protein as bait. Each strain was re-coated on SD/-Trp/-Leu/-Ade/-His+x-α-gal plates, and 45 genes failed to grow or produced only white dots on the plates. The 53 monoclonal lines that could grow and showed blue spots were tested via PCR amplification. In total, fifteen bacterial fluid bands were amplified (Appendix A). These 15 bands were counted, and 10 bands of different sizes were found. These *E. coli* fluids were further sequenced to obtain seven corresponding genes. A comparison of the CDS sequences of the seven genes in NCBI revealed that most of the genes were closely related to drought stress, including 26S proteasome regulatory subunit 4 homolog A-like, probable Adenosine Diphosphate(ADP)-ribosylation factor Guanosine Triphosphate(GTP)ase-activating protein *Arabidopsis* G-protein Signaling 13(AGD13)-like, sucrose non-fermenting 1(SNF1)-related protein kinase catalytic subunit alpha KIN10, basic leucine zipper(bZIP), NIGHT LIGHT-INDUCIBLE AND CLOCK-REGULATED(LNK)2, transcription factor EMBRYO DEFECTIVE (EMB1444), and 40S ribosomal protein S20-2-like.

### 2.5. Validation of VvSnRK2.7 Interactions with bZIP Protein

To study the relevant interaction proteins of *VvSnRK2.7*, seven genes were screened from the grape yeast library using *VvSnRK2.7* as bait. The pGADT7 recombinant vector was constructed for each of the seven genes, co-transformed with pGBKT7-*VvSnRK2.7* in Y2Hgold yeast receptor cells, and verified using pGADT7/pGBKT7-53m and pGADT7/pGBKT7-Lam as positive and negative controls, respectively. The cotransformed yeast cells were coated on plates of SD/-Trp/-Leu, SD/-Trp/-Leu+ x-a-gal and SD/-Trp/-Leu/-Adei-His, and SD/-Trpi-Leu/-Adei-His+ x-a-gal. We found that yeast cells co-transfected with pGBKT7-*VvSnRK2.7* and pGADT7-VvbZIP recombinant plasmids and positive control yeast cells grew blue lines on the plates. Negative control yeast cells grew and presented only a white color on the two deficient plates but did not grow on the four deficient plates. This result indicates that pGBKT7-*VvSnRK2.7* and pGADT7-VvbZIP proteins have strong interactions (Figure 4).

### 2.6. pCAMBIA1300-VvSnRK2.7 Overexpression Vector Construction and Characterization of Transgenic Arabidopsis

The target fragment of *VvSnRK2.7* was ligated to the overexpression vector pCAMBIA1300. Next, the recombinant plasmid pCAMBIA1300-*VvSnRK2.7* was introduced into *A. tumefaciens* GV3101 via the flocculation dip method to obtain transgenic *Arabidopsis*. The T3 generation of the pure lines was obtained after two screenings (Appendix A #2, #5, and #7), and the extracted DNA of transgenic lines was detected via gel electrophoresis using a DNA Extraction Using Kit, as shown in Appendix A. The detected sizes of electrophoretic bands were consistent with those of the target gene fragments. The lines (#2, #5, and #7) were used for subsequent functional verification tests.

### 2.7. Morphological Observation and Expression Analysis of Transgenic Arabidopsis after Drought Stress

*Arabidopsis* overexpression lines grown for 4 weeks were treated with 300 mmol·L^−1^ mannitol and photographed for observation at 0, 3, and 5 d (Figure 5). No phenotypic differences were observed between the overexpressed *Arabidopsis* lines and the WT at 0 d. At 3 d, both the WT *Arabidopsis* and overexpressed *Arabidopsis* lines showed yellowing at the leaf margins, with a higher degree of yellowing in the WT than in the transgenic lines. At 5 d, leaves of wild-type *Arabidopsis* yellowed and dried. In contrast, yellowing increased in the transgenic strain, but no desiccation occurred. We hypothesized that the transgenic lines were more drought resistant than those of the WT *Arabidopsis*.

### 2.8. VvSnRK2.7 Increased Antioxidant Enzyme Activities and Proline Content under Drought Stress

Under drought stress, dysregulation of plant reactive oxygen species (ROS) is induced, and the oxidase system plays an important role in scavenging ROS from the plant. In addition, proline, which acts as a store of nitrogen and energy, can provide energy for a number of energy-consuming physiological processes, especially during recovery from adverse conditions. Therefore, in the present experiment, WT and transgenic *Arabidopsis* were subjected to drought stress treatment using 300 mmol·L^−1^ mannitol, and fresh samples were taken at 0 d, 3 d, and 5 d to determine the contents of POD, SOD, CAT, and Pro in *Arabidopsis* leaves. The results (Figure 6) indicate that the contents of POD, SOD, CAT, and Pro in both wild-type (WT) and transgenic lines of *Arabidopsis* increased with an increase in drought stress time. Moreover, the contents of POD, SOD, CAT, and Pro in the leaves of transgenic lines were significantly higher than those of the wild type at 3 d and 5 d, indicating that the overexpression of *VvSnRK2*.7 enhances drought resistance in *Arabidopsis*.

### 2.9. Overexpression of the VvSnRK2.7 Alters Arabidopsis Leaf MDA Content and Relative Conductance

To understand whether *VvSnRK2.7* reduces the osmotic action of cell membranes under drought stress, the MDA content and relative conductivity of WT *Arabidopsis* and transgenic *Arabidopsis* were determined under different drought stress durations. The results are shown in Figure 7, which shows that in the absence of drought stress, there was no significant difference in the conductivity of MDA content between the WT and transgenic lines. However, under prolonged stress duration, the MDA content and relative conductivity of both WT *Arabidopsis* and transgenic *Arabidopsis* lines were elevated, with a greater magnitude of elevation in the WT than in the transgenic lines. Moreover, the WT *Arabidopsis* had a significantly higher MDA content and relative conductivity than those of the transgenic lines. Therefore, it can be concluded that overexpression of the *VvSnRK2.7* in *Arabidopsis* can slow the effects of drought stress on cell membrane permeability and ensure normal functioning of the intracellular environment.

### 2.10. Overexpression of VvSnRK2.7 Increases Leaf Sugar Content in Arabidopsis

We used high-performance liquid chromatography to measure the contents of fructose, glucose, and sucrose in *Arabidopsis* leaves to determine whether overexpression of *VvSnRK2.7* can alter sugar content. The results are shown in Figure 8, which shows no significant difference in the fructose, glucose, and sucrose contents in the leaves of WT *Arabidopsis* and transgenic *Arabidopsis* lines in the absence of drought stress. After three days of drought stress, the contents of fructose, glucose, and sucrose in the leaves of transgenic *Arabidopsis* were significantly higher than those of the wild type, with the highest content of fructose in OE3 (1.262 mg·g^−1^) (68 percent). OE7 had the highest glucose and sucrose content (9.474 and 1.257 mg·g^−1^, respectively), representing values 60.7 and 81.1 percent higher, respectively, than those of WT *Arabidopsis*. After 5 d of stress, the fructose, glucose, and sucrose contents of WT *Arabidopsis* were significantly lower than those of transgenic *Arabidopsis*. The fructose content of WT *Arabidopsis* leaves was 0.9 mg·g^−1^, the glucose content was 8.23 mg·g^−1^, and the sucrose content was 0.610 mg·g^−1^. The fructose content of leaves from transgenic *Arabidopsis* lines was 1.21~1.47 mg·g^−1^, the glucose content was 12.508~13.230 mg·g^−1^, and the sucrose content was 0.813~0.894 mg·g^−1^. Thus, overexpression of *VvSnRK2.7* significantly increased the fructose, glucose, and sucrose contents of *Arabidopsis* leaves under drought stress.

## 3. Discussion

*SnRK2* plays an important role in the ABA signaling transcription pathway [19]. ABA promotes the expression of sugar transporter proteins in berries, which facilitates the unloading of sugar from the phloem into the reservoir cells (berries) [20]. In order to clarify the location of *VvSnRK2.7* expression in cells, we observed the green fluorescent protein signal by utilizing the A. tumefaciens-mediated method to inject *VvSnRK2.7* into the leaves of Benjamin tobacco. Researchers previously found that the *N. benthamiana* NtSnRK2.9-11 protein in group 3 is present in the nucleus, cell membrane, and cytoplasm [1]. Conversely, in this assay, *VvSnRK2.7* was localized in the nucleus. This discrepancy may be due to varietal and genetic differences (Figure 3). In addition, the oat (*Avena sativa* L.) AsSnRK2.7 protein was previously found to be localized in the nucleus [21], which is similar to the results of the present study.

To further investigate the physiological mechanism by which *VvSnRK2.7* regulates sugar metabolism under drought stress, the present study utilized a yeast two-hybrid assay to screen for VvbZIP-interacting proteins (Figure 4). Studies have shown that the expression of grape *VvZIP23* is strongly induced by drought, low temperature, and high salt [22]. Tissue specificity further suggests that *bZIP* is involved in the growth and developmental processes of grape organs [23]. Therefore, the *VvbZIP* transcription factor derived from the yeast two-hybrid screen in this experiment has an important role in enhancing plant drought tolerance and improving photosynthetic product metabolism under drought stress.

Grapes are subjected to abiotic stresses such as high salt, drought, and low temperature during their growth and development. These stresses adversely affect the formation of fruit quality. Previous studies found that the *SnRK2* gene regulates sugar metabolism under drought stress [24] and enhances drought tolerance in plants [25]. Transcriptome analysis in this experiment also indicated that *SnRK2* is associated with sugar metabolism and drought. To further explore its functions, *VvSnRK2.7* was transfected into *Arabidopsis* for overexpression. Subsequently, we explored the mechanism by which it regulates berry sugar content under drought stress as a way to verify the gene function of *VvSnRK2.7*.

We found that the *SnRK2* gene plays an important role in plant responses to osmotic stress and that overexpression of the *SnRK2* gene significantly improved drought tolerance in plants. For example, overexpression of wheat *TaSnRK2*.7 in *Arabidopsis* improved drought tolerance in *Arabidopsis* [26]. In addition, overexpression of the *Arabidopsis AtSnRK2*.8 gene significantly increased drought tolerance in *Arabidopsis* seedlings [25], whereas mutants of AtSnRK2.1-10 were significantly less resistant to hyperosmotic stress [27]. The transgenic *Arabidopsis* overexpressing *VvSnRK2.7* in this experiment was not significantly different from the wild type when not being subjected to drought stress. The expression of *VvSnRK2.7* in the transgenic lines was significantly higher than that in the wild type when the transgenic lines were treated with wild-type *Arabidopsis* for 3 d and 5 d of drought stress (Figure 5). Relative conductivity and MDA measurements of the transgenic lines and WT *Arabidopsis* revealed that the conductivity and MDA content of WT and transgenic *Arabidopsis* not subjected to drought stress were 20~25% and 25~35 nmol-g^−1^, with no significant difference between the WT and transgenic lines of *Arabidopsis*. When drought stress treatment was applied for 3 d and 5 d, the conductivity and MDA content of both the WT and transgenic *Arabidopsis* increased. However, during the same period, the conductivity and MDA content of the WT plants were significantly higher than those of the transgenic lines. Specifically, overexpression of *VvSnRK2.7* significantly reduced the relative conductivity and MDA content of *Arabidopsis* and better maintained the stability of the cellular environment. Ultimately, the results of the present experiments were similar to those in previous studies (Figure 7).

The *SnRK2* gene not only plays an important role in plant drought response but also participates in plant sugar signaling and sugar metabolism [13,28]. At the same time, overexpression of *TaSnRK2*.8 significantly increases the soluble sugar content under drought conditions [24]. In this experiment, the sucrose, glucose, and fructose contents of *Arabidopsis* leaves overexpressing *VvSnRK2.7* were not significantly different from those of WT *Arabidopsis* leaves at 0 d. Conversely, the amount of each sugar was elevated at 3 d and 5 d of drought stress compared with the previous period, and the amount of each sugar in the WT *Arabidopsis* was significantly lower than that in the transgenic *Arabidopsis* lines (Figure 8). Ultimately, overexpression of *VvSnRK2.7* increased the sugar content of *Arabidopsis* leaves under drought stress, similar to the results of the previous study.

## 4. Materials and Methods

### 4.1. Test Materials

Colombian WT *Arabidopsis* was used to analyze the overexpression of *VvSnRK2.7.* The transgenic *Arabidopsis* seeds were filtrated through an MS medium supplemented with 50 mg·L^−1^ kanamycin and 300 mg·L^−1^ timentin. Then, the positive lines were grown in medium containing the matrix and vermiculite (3:1) and remained there for 16 h at 25 °C and 8 h at 22 °C (light/dark) in a growth chamber. The transgenic and WT *Arabidopsis* were treated with 300 mmol·L^−1^ mannitol at 5 d.

*N. benthamiana* was used for the subcellular localization analysis of the VvSnRK2.7 protein. The *N. benthamiana* was cultivated in a growth chamber with a cycle of 16 h light/8 h dark/light at 22/18 °C with a light intensity of 10,000 lx. After four weeks, the resuspension solution was injected into the tobacco leaf abaxial epidermis using a needleless syringe.

### 4.2. Obtaining and Cloning of Grape VvSnRK2.7

*VvSnRK2.7* (*GSVIVG01022427001*) was screened using transcriptome analysis [29], and its CDS fragment was downloaded from the grape genome website (https://www.genoscope.cns.fr/externe/GenomeBrowser/Vitis/, accessed on 8 March 2023), with a length of 1068 bp. Through homologous recombination, the amplification primers were designed using the CE Design V1.04 software with BamHⅠ as the enzyme cleavage site. The primer sequences are shown in Table 2. 

### 4.3. Overexpression and Subcellular Localization of Recombinant Plasmid Transformation

The *A. tumefaciens*-mediated genetic transformation of *Arabidopsis* was induced via inflorescence dipping [30]. For this process, 4-week-old *Arabidopsis* was regarded as the experimental material and infected using the floral-dipping method. Overexpressed recombinant plasmid pART-CAM-VvSnRK2.7-EGFP was thus transformed into *Arabidopsis* (Ecotype Columbia). *A. tumefaciens* GV3101 carrying VvSnRK2.7 was grown in LB media with 100 mg·L^−1^ kanamycin and 100 mg·L^−1^ rifampicin. We collected and suspended A. tumefaciens during the log phase of growth (OD_600_ = 0.4–0.6) in an MS medium supplemented with 5.0% (wt/vol) sucrose, 100 μmol·L^−1^ acetosyringone (AS), and 0.05% (vol/vol) Silwet. The infection time remained 2 min throughout the process. The infected experimental materials were placed in the dark for 24 h and then transferred to normal conditions. Transgenic *Arabidopsis* was identified by PCR using overexpression-specific primers for *VvSnRK2.7*.

Transient expression assays were employed to transiently transform tobacco leaves [31]. First, the open reading frame (ORF) of VvSnRK2.7 was amplified using gene-specific primers to assess subcellular localization (Table 1). Then, we attached the frame to the 5′ end of the reporter gene GFP (green fluorescence protein gene), which was inserted into the plasmid pART-CAM-EGFP to generate a combined VvSnRK2.7-EGFP. Lastly, this combination was used to induce the transient transformation of tobacco. The protoplasts were observed using a laser scanning confocal microscope (Olympus FV1000 viewer, by Olympus Company, Tokyo City, Japan) under weak light for 12–18 h.

### 4.4. Functional Analysis of Drought Resistance in Transgenic Arabidopsis

Four-week-old *Arabidopsis* plants were subjected to mannitol-based simulated drought stress. The treatments employed 300 mmol·L^−1^ mannitol, and the control used an equal volume of distilled water. Samples were taken after 0 d, 3 d, and 5 d of simulated drought stress to determine the physiological indices and quantify the genes. The overexpressed VvSnRK2.7 was infused into the pART-CAM-EGFP vector.

### 4.5. Measurement of Relevant Indicators after Stress

Conductivity was determined using the method of Rohde et al. [32]. Thawing leaves were immersed in 7 mL of distilled water and placed on a shaker for 16 h at 4 °C. Electrolyte leakage was determined as the ratio of conductivity measured in the water before and after boiling the samples using a DDS-307A conductivity meter by INESA Scientific Instrument Co., Ltd., Shanghai City, China. MDA, SOD, POD, and CAT contents were determined using the instruction manual of the Suzhou Comin Biochemical Kit. Sugar content was determined using the method of Ma et al. [29]. Fructose, glucose, and sucrose contents of *Arabidopsis* leaves were determined via Waters Acquity Arc high-performance liquid chromatography (HPLC) using by Waters Company, Milford City, MA, USA.

An amount of 0.5 g of the sample was accurately weighed and extracted by grinding with 5 mL of 80% ethanol and centrifuged at 12,000 r/min for 15 min at 4 °C; then, the supernatant was taken. The extraction was repeated twice. Each time, 2 mL of 80% ethanol was added, the supernatant was combined, and the volume was increased to 10 mL. The sample was placed in vacuum centrifugal concentrator rotary evaporation (60 °C), and rotary evaporation was performed for 3 h to dry it, with 1 mL of ultrapure water and 1 mL of acetonitrile to make the sample re-soluble. Then, it was filtered through a 0.22 μm microporous membrane of the organic phase, and the filtrate was added to the sample vials to be measured.

The separation was carried out on a Waters Acquity Arc high-performance liquid chromatograph (HPLC) with a column (4.6 mm × 150 mm, 2.5 µm) at 40 °C with the following: mobile phase 75% acetonitrile + 0.2% triethylamine + 24.8% ultrapure water; flow rate 0.8 mL/min; injection volume 10 μL. The standard samples were chromatographically pure sucrose, fructose, and glucose supplied by Sigma Company, St. Louis, MO, USA. The standard curve was prepared by mixing different concentrations of the standards.

### 4.6. Screening of VvSnRK2.7-Interacting Proteins with a Mating Assay

After the collection of leaves of ‘Pinot Noir’ grape plants after drought stress treatment, total RNA was extracted from the collected samples using the CTAB method, and the mRNA was isolated from the extracted total RNA using Fast Track MAG beads. ThemRNA was reversely transcribed to create double-stranded cDNA, which was then ligated to the attB1 recombinant junction. The primary library plasmid obtained above was used for an LR reaction using LR Clonase II Mix and connected to the PGADE-T7 vector. Finally, a yeast two-hybrid library plasmid was obtained.

The Y2Hgold yeast sentient cell preparation and Mating hybridization screening for interaction proteins used the method of Ren et al. [33]. The Mating method involved placing the yeast strain Y2Hgold containing the recombinant plasmid pGBKT7-*VvSnRK2.7* into 1 mL of SD/-Trp liquid medium followed by incubation overnight. Next, 500 μL of the bacterial solution was mixed with 50 mL of the SD/-Trp liquid medium to achieve an OD_600_ = 0.8, followed by using the Mating method to screen for interacting proteins.

### 4.7. Validation of the VvSnRK2.7 with VvbZIP

A cDNA library of Pinot Noir grape responses to drought was constructed in the laboratory. The pGADT7 and pGBKT7 vectors were digested with BamHⅠ, pGBKT7-VvSnRK2.7, and pGADT7-VvbZIP primers used to amplify ORF (Table 1). Then, pGADT7-VvbZIP and pGBKT7-VvSnRK2.7 plasmids were co-transformed into Y2Hgold yeast receptor cells. Next, 10 μL of the recombinant plasmid pGBKT7-VvSnRK2.7 and 3 μL of Carrier DNA were added to two tubes of the yeast receptor cells prepared above. The first time the Carrier DNA was used, it was boiled in water for 20 min and then immediately inserted into ice and stored at −20 °C. Then, 600 μL of a PEG/LiAc solution was added to each tube with vigorous shaking. Afterwards, the tubes were incubated for 30 min at 30 °C and 200 rpm in a constant temperature shaker. After shaking, 70 μL of Dimethyl sulfoxide was added to each tube, mixed upside down, and heat-excited at 42 °C for 15 min, with five-minute intervals of mixing in a vertical and horizontal orientation to improve conversion efficiency. At the end of the heat treatment, the centrifuge tubes were quickly inserted into ice for 5 min. The tubes were centrifuged at 25 °C at 14,000 rpm for 1 min, and the supernatant was aspirated as much as possible with a pipette gun. Then, the yeast cells were resuspended with 0.1 mL of 1× TE or sterilized water. Next, the two tubes of suspended cells were spread on SD/-Trp/-Leu and incubated in an inverted culture at 30 °C for 2 d. Lastly, the interactions were validated using SD/-Trp/-Leu/-His/-Ade+x-α-gal.

### 4.8. qRT-PCR Analysis

*Arabidopsis* leaf RNA was extracted using the CTAB method. RNA quality and quantity were determined using a Pultton P200 Micro Volume Spectrophotometer (Pultton Technology Ltd., San Jose, CA, USA). RNA was stored at −80 °C for further analysis.

The primers (Table 1) were synthesized by Shanghai (Shanghai, China) Biological Engineering Co., Ltd. RNA was extracted from transgenic and WT *Arabidopsis* and reverse transcribed with single-stranded cDNA as the template. Atactin2 was used as the internal reference gene, and the quantitative reaction system was 20 µL, as follows: 1 µL cDNA, 1 µL each of upstream and downstream primers (10 µmol/L), 10 µL SYBR enzyme, and 7 µL ddH_2_O. The reaction procedure was as follows: 95 °C predenaturation for 30 s, 95 °C denaturation for 10 s, 60 °C annealing for 30 s, and 72 °C extension for 30 s (40 cycles). The test was repeated 3 times. Then, the reaction procedure, melting curve, and fluorescence value change curve were analyzed.

### 4.9. Data Statistical Analysis

A statistical analysis of the VvSnRK2.7 expression data was performed using SPSS 22.0, and Duncan’s multiple range tests were employed to test significant differences (*p* < 0.05).

## 5. Conclusions

Previous screening using transcriptomic data and subcellular localization revealed that *VvSnRK2.7* was localized in the cell membrane and nucleus. The yeast two-hybrid screen showed that the VvSnRK2.7 protein interacts with the VvbZIP protein and that VvbZIP may activate VvSnRK2.7 expression. In addition, heterologous expression of *VvSnRK2.7* in *Arabidopsis* revealed that *VvSnRK2.7* increased antioxidant enzyme activities, as well as the proline, sucrose, glucose, and fructose content, and decreased the conductivity and MDA content. These results demonstrate that *VvSnRK2.7* can maintain cellular structural stability, reduce cellular osmotic patterns, and improve plant drought resistance.

## Figures and Tables

**Figure 1 ijms-25-04473-f001:**
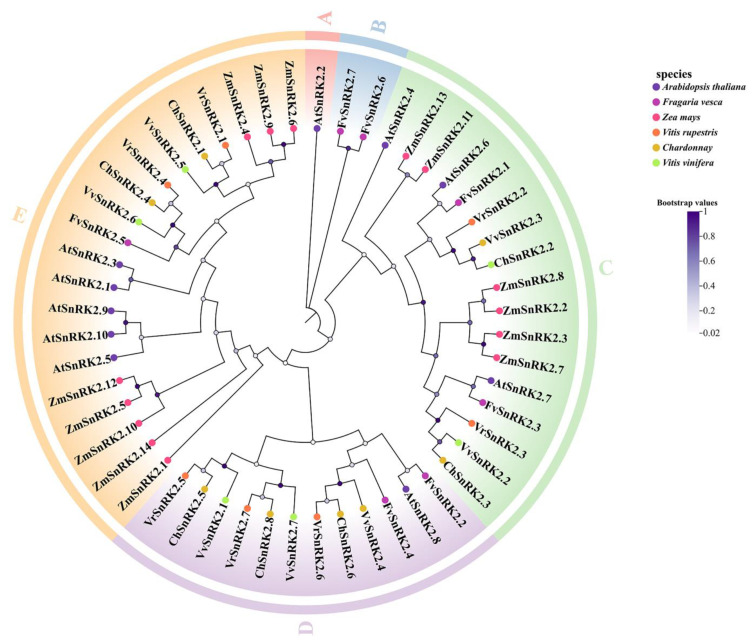
Evolution analysis of the *VvSnRK2* gene in grape. The NJ method was adopted, and the bootstrap value was set to 1000. Six species were selected for phylogenetic tree analysis. At: *Arabidopsis*; Fv: *Fragaria vesca*; Zm: *Zea mays*; Vr: *Vitis rupestris*; Ch: *Chardonnay*; Vv: *Vitis vinifera*. A–E: Different subfamilies of SnRK2 gene in *Arabidopsis thaliana*, *Fragaria vesca*, *Zea mays*, *Vitis rupestris*, *Chardonnay*, and *Vitis vinifera*, respectively.

**Figure 2 ijms-25-04473-f002:**
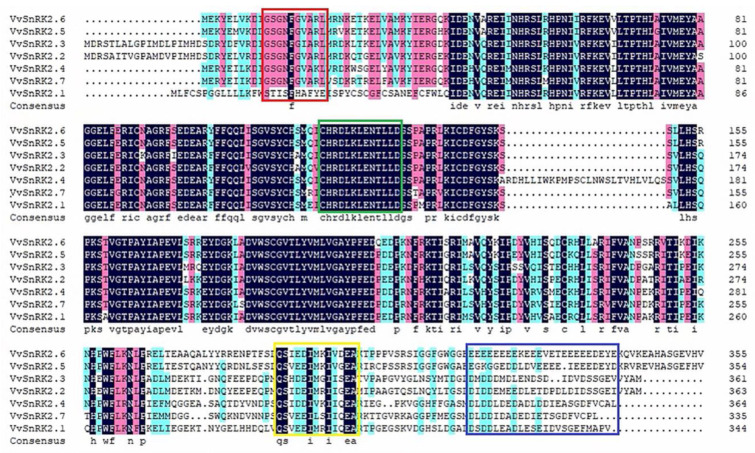
Multiple sequence alignment of seven VvSnRK2 proteins with conserved structural domains indicated by red, green, yellow, and blue boxes. The red box represents the ATP-binding domain, the green box represents the active site of the serine/threonine protein kinase, the yellow box represents the abiotic stress response domain, and the blue box represents the ABA-inducible regulatory domain. Different colors represent highlight homology level, where black: ≥100%, pink: ≥75%, light blue: ≥50%, yellow: ≥33%.

**Figure 3 ijms-25-04473-f003:**
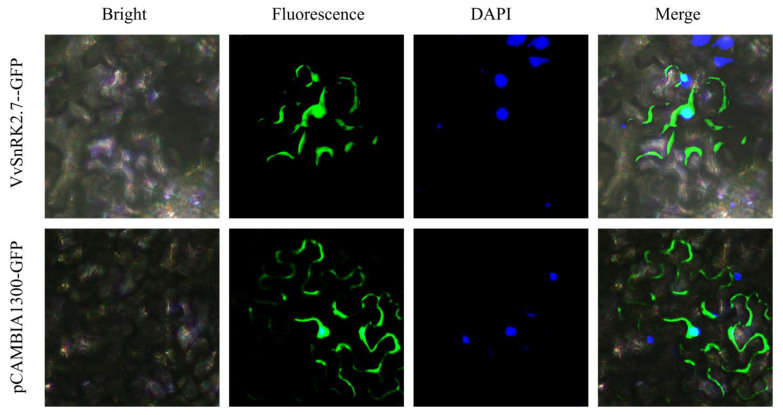
Localization of *VvSnRK2.7* in tobacco cells. The control vector (35S::GFP) and 35S::VvSnRK2.7::GFP were transformed in *Nicotiana benthamiana* (*N. benthamiana*) leaves. The GFP signals in cells were observed using confocal microscopy.

**Figure 4 ijms-25-04473-f004:**
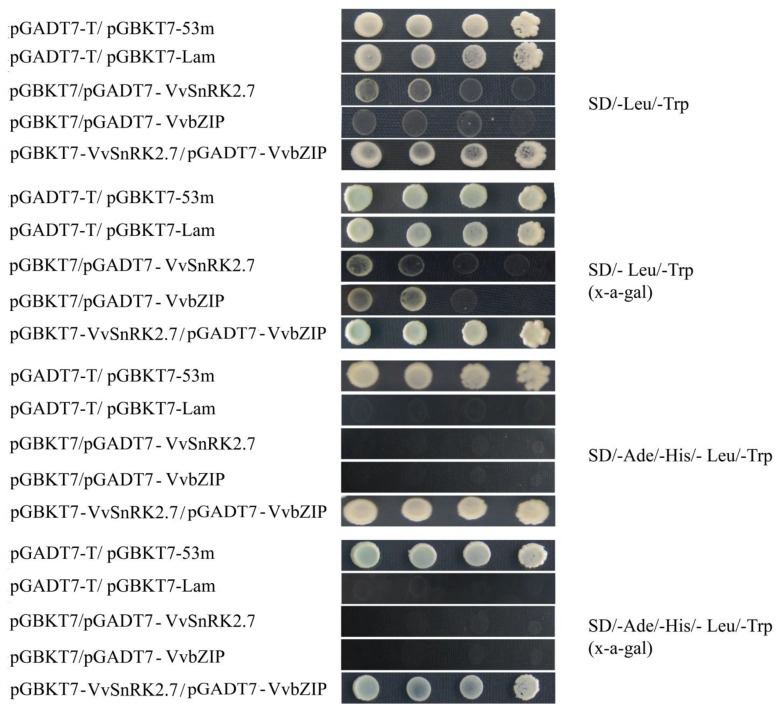
Interaction verification between pGBKT7-VvSnRK2.7 and pGADT7-VvbZIP protein yeast. AD and BK represent pGADT7 and pGBKT7, respectively.

**Figure 5 ijms-25-04473-f005:**
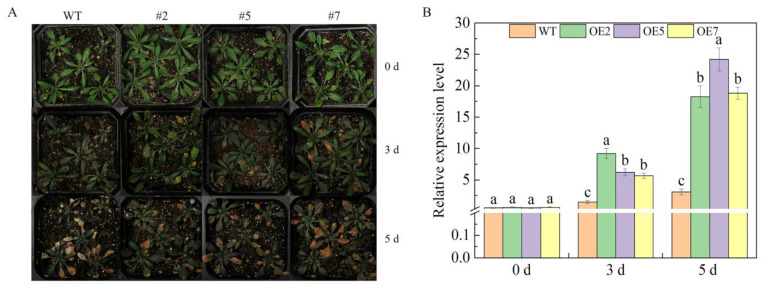
Phenotypic observation and gene expression of *VvSnRK2.7*-overexpressed *Arabidopsis* lines under drought stress. (**A**) Phenotypes of *Arabidopsis* under drought treatment. Drought stress was simulated with 300 mmol-L^−1^ mannitol, and controls used equal volumes of distilled water, WT (WT strain), #2, #5, and #7 (transgenic strain); 0 d, 3 d, and 5 d (days of simulated drought stress treatment). (**B**) Expression of *VvSnRK2.7* in *Arabidopsis* under the effect of simulated drought. The relative expression of genes was calculated via the 2^−∆∆Ct^ method, and statistical significance was analyzed with a one-way ANOVA. Error bars represent the mean ± SE from three biological repeats. Different letters denote significant differences, whereas the same lowercase letters indicate no statistical difference (*p* < 0.05).

**Figure 6 ijms-25-04473-f006:**
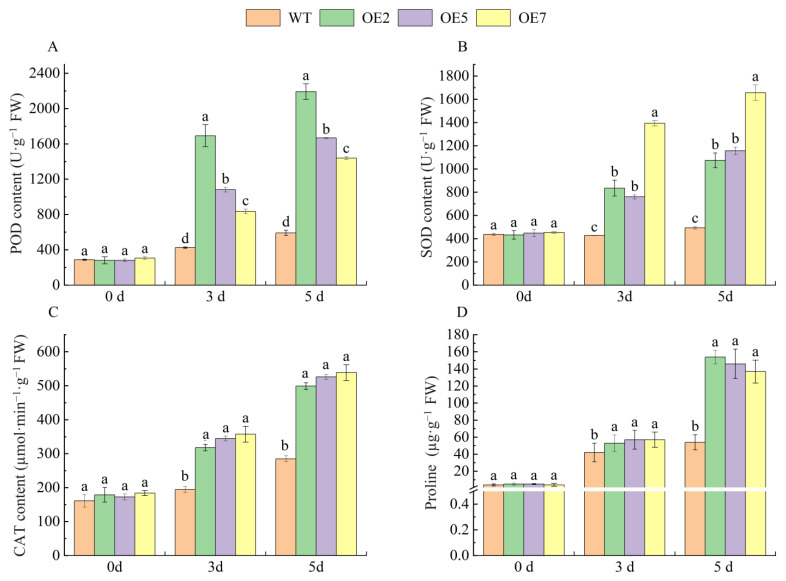
Antioxidant enzyme activity and proline content of *VvSnRK2.7* transgenic *Arabidopsis* under drought stress. (**A**) Peroxidase (POD) content of *VvSnRK2.7* transgenic *Arabidopsis* under drought stress. (**B**) Superoxide dismutase (SOD) content of *VvSnRK2.7* transgenic *Arabidopsis* under drought stress. (**C**) Catalase (CAT) content of *VvSnRK2.7* transgenic *Arabidopsis* under drought stress. (**D**) Proline content of *VvSnRK2.7* transgenic *Arabidopsis* under drought stress. The relative expression of genes was calculated using the 2^−∆∆Ct^ method, and statistical significance was analyzed using one-way ANOVA. Error bars represent the mean ± SE from three biological repeats. Different letters denote significant differences, whereas the same lowercase letters indicate no statistical difference (*p* < 0.05).

**Figure 7 ijms-25-04473-f007:**
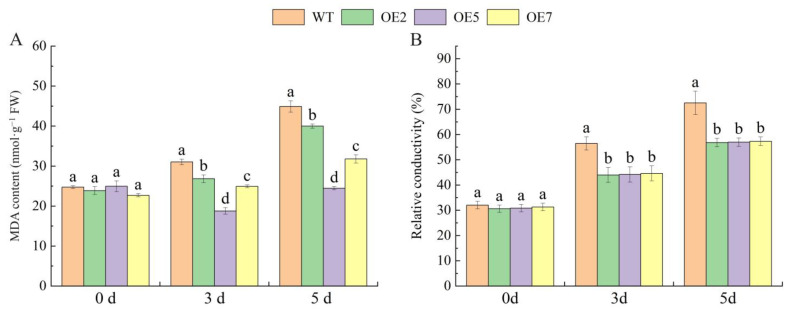
Influence of *VvSnRK2.7* on membrane permeability. (**A**) Malondialdehyde (MDA) content of *VvSnRK2.7* transgenic *Arabidopsis* under simulated drought stress. (**B**) Relative conductivity of *VvSnRK2.7* transgenic *Arabidopsis* under simulated drought stress. The relative expression of genes was calculated with the 2^−∆∆Ct^ method, and statistical significance was analyzed via one-way ANOVA. Error bars represent the mean ± SE from three biological repeats. Different lowercase letters denote significant differences, whereas the same lowercase letters indicate no statistical difference (*p* < 0.05).

**Figure 8 ijms-25-04473-f008:**
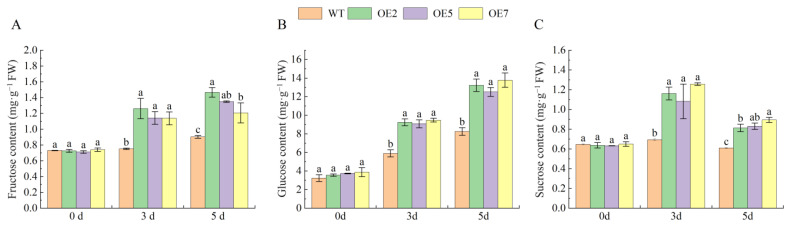
Effects of *VvSnRK2.7* on sugar content in *Arabidopsis* leaves. *(***A**) Fructose content of *Arabidopsis* leaves. (**B**) Glucose content of *Arabidopsis* leaves. (**C**) Sucrose content of *Arabidopsis* leaves. Different small letters indicate significant difference at *p* < 0.05.

**Table 1 ijms-25-04473-t001:** Physicochemical properties and subcellular localization analysis of the VvSnRK2 protein.

Gene	Amino Acid Number	Molecular Weight	Isoelectric Point	Full Length/bp	Instability Index
*PNVvSnRK2.7*	355	40.61	6.05	5680	45.07
*RGVvSnRK2.1*	355	40.62	6.1	5683	47.12
*ChVvSnRk2.1*	355	40.68	6.05	5690	47.12

**Table 2 ijms-25-04473-t002:** Gene amplification and primers detected by bacterial liquid PCR.

Gene	Primer Sequence (5′-3′)	Purpose
*VvSnRK2.7*-EGFP	5′-gagctcggtacccggggatccATGGAGAAGTATGAAATGGTGAAGG	Transient expression
	3′-ggtgtcgactctagaggatccACTAACGTGAAATTCTCCGCTTG	Transient expression
pCAMBIA1300-*VvSnRK2.7*	5′-gagctcggtacccggggatccATGGAGAAGTATGAAATGGTGAAGG	Overexpression
	3′-ggtgtcgactctagaggatccTTAACTAACGTGAAATTCTCCGCT	Overexpression
pGBKT7-*VvSnRK2.7*	5′-aggccgaattcccggggatccttATGGAGAAGTATGAAATGGTGAAGG	BD vector primers
	3′-ccgctgcaggtcgacggatccTTAACTAACGTGAAATTCTCCGCT	BD vector primers
pGADT7-*VvbZIP*	5′-gtgggcatcgatacgggatccatATGTTGTCATCAACAGGTGGCG	AD vector primers
	3′-cagctcgagctcgatggatccTCAAAATGGGGCTGTTGAAGTT	AD vector primers
qRT-*AtActin*	5′-CTTGCACCAAGCAGCATGAA	qRT-PCR primers
	3′-CCGATCCAGACACTGTACTTCCTT	qRT-PCR primers
qRT-*VvSnRK2.7*	5′-AGAAGAGGGTAAAGGCGGTGAGG	qRT-PCR primers
	3′-R-CGCTTGCATGGACTTCCCTAACC	qRT-PCR primers

Lowercase letters in the table are homology arms and uppercase letters are primers.

## Data Availability

Data will be made available on request.

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
