# Peer review of "Grape SnRK2.7 Positively Regulates Drought Tolerance in Transgenic *Arabidopsis"

_ijms, 2024, doi:10.3390/ijms25084473_

Round 1

Reviewer 1 Report

Comments and Suggestions for Authors

The manuscript “Grape SnRK2.7 positively regulates the drought tolerance in transgenic Arabidopsis” investigated the role of VvSnRK2.7 gene in grapevine under stress. The authors have overexpressed VvSnRK2.7 in Arabidopsis and analyzed the expressions and the antioxidant enzyme activity, MDA and proline content. Overall, the manuscript has the potential to bring useful information for both grape breeders and plant biologists in general. However, I have some major concerns before its acceptance in the IJMS journal.

1.    Abstract have written in a redundant manner. All the methods that have been described here are not necessary as the author explains in the method section. The method should be more concise and major findings should be included here only.

2.    In line 14, “Arabidopsis strain” should be “Arabidopsis lines”

3.    In line 43-44, author only stated that SnRK2.7 gene selected from the transcriptome data of 'Guirenxiang' grapes drought resistance research analyze the expression of all SnRK2 genes but did not provide the data. The author must need to provide the expression data of all the SnRK2 genes and need to clarify why they have selected SnRK2.7 gene instead of others.

4.    In line 45-46, “which is beneficial to It is beneficial to investigate the regulatory mechanism of SnRK2.7 in response to drought stress”. Sentence need to be corrected.

5.    In line 43, author stated that SnRK2.7 gene selected from the transcriptome data of 'Guirenxiang' grapes drought resistance research, but in line 75, The VvSnRK2.7 gene was screened from the transcriptome data of 'ltalian Riesling' grapes in the previous study. It is confusing from which data they have selected VvSnRK2.7 gene? Author must need to provide the transcriptome data for all the VvSnRK2 genes and need to clarify why they have selected SnRK2.7 gene instead of others.

6.    In Figure 2 legend, although conserved structural domains indicated by red, green, yellow and blue boxes. Which color box does it stand for which needs to specify?

7.    In line 115, Agrobacterium tumefaciens should be in Italic.

8.    In line 161, ‘Strains’ should be ‘lines’

9.    In line 196, Arabidopsis should be Italic

10.  Figure 8 was not provided in the manuscript.

11. In 253-260, Author identified VvbZIP as the interaction partner of VvSnRK2.7 and discuss the importance of VvbZIP gene under drought stress, but I am concern how VvSnRK2.7 regulate the VvbZIP gene? Author needs to discuss here for the general reader. Did the author check the expression of Arabidopsis bZIP gene in the VvSnRK2.7 transgenic plants? Any changes in VvSnRK2.7 transgenic plants?

12. In line 357, VvSnRK2.7 was localized in the cell membrane and nucleus not consistent with previously described in the result section.

13. Throughout the manuscript, sentence making and English are confusing some mention above, author must need to revise appropriately to understand their results.

14. Referencing is not correct throughout the manuscript. Must need to revise according to journal guideline.

15. Supplementary figures don’t have any figure legends. Need to revise.

Comments on the Quality of English Language

Moderate editing of English language required. Some I have mentioned in the comment section.

Reviewer 2 Report

Comments and Suggestions for Authors

Manuscript ijms-2938882 “Grape SnRK2.7 positively regulates the drought tolerance in transgenic Arabidopsis”. This manuscript is devoted to studying the functions of a protein kinase SnRK2.7 from grape Vitis vinifera. While plant protein kinases are implicated in a variety of processes and posses pivotal regulatory functions in plant cells, there is scarce information on SnRK in plants. This manuscript provides new data on SnRK functions in plant cells using modern and valuable methods, such as protein target identification using yeast two hybrid screening. The topic of this manuscript is interesting and is within the scope of IJMS. It appears that this manuscript is well done using modern approaches but need extensive corrections to improve data presentation, matherial and methods section, and language. There are a number of important issues that should be addressed before further processing.

Specific edits:

1) Introduction should be better divided into paragraphs. Paragraphs one and two are too large. Please divide the paragraphs into 2-3 ones. The last paragraph of the Introduction should inform briefly what has been done in this study. Also, it is necessary to clearly mention the aim of the study in the Introduction.

2) Figure legends should be considerably improved and extended.

- Figure 5. The title of the figure legend includes “determination of enzyme activities” . However, there are no enzyme activities on the Figure. Did you mean SnRK gene expression?

- The Figure 5 title title should also include a brief description of how drought stress was applied and mention the method used for gene expression analysis.

- Statistical analysis explanation should also be included in the legend, i.e. mean±standard error or deviation? What type of statistical significance analysis has been done, i.e. Student’s t-test or ANOVA and what type? Explain also the letters “a”, “b”,.. in the legend.

- Figure 6 and Figure 7. Statistical analysis explanation should also be included in the legend, i.e. mean±standard error or deviation? What type of statistical significance analysis has been done, i.e. Student’s t-test or ANOVA and what type?

Explain in the Figure legends the abbreviations POD, SOD, CAT, MDA.

 3) “Material and Methods” section is too brief and lacks basic information that is necessary to understand how the work has been done.  M&M should be considerably extended. I recommend the following specific edits:

- 4.1. Test materials. Provide description how the plants were grown, the growth conditions, temperature, light, cultivation chamber, etc. How the seeds have been germinated?

- Section 4.2. “from the grapevine genome website”. Please provide the website name and address.

- Section. 4.4. How drought stress has been applied? Include a more detailed explanation here. Did you mean by mannitol application? If simple mannitol application, then this is not drought stress, mannitol application is used to induce osmotic stress. Drought stress is more than just osmotic stress and it should include water deficit induced without mannitol. Did you apply drought by withholding water? If not, why not? Anyway, a detailed description of stress application should be included.

Section 4.3 and Section 4.5. Brief explanations of the methods used should be included (not just paper references).  At least, one-two sentences to explain each mentioned method or provide some key method details.

Section 4.7  “…using grape cDNA as a template”. Please explain in this section clearly what grape cDNA did you use and how did you obtain it? How did you purify RNA and construct cDNA library? Or you bought commercial grape cDNA library? All details should be provided, probably in a separate M&M subsection.

Please also explain briefly the Mating method? Also, is it really should be mentioned as “Mating method” not “mating method”?

- “After that,  VvbZIP-pGADT7  and VvSnRK2.7-pGBKT7 recombinant plasmids were co-transformed into Y2Hgold yeast receptor cells.” Explain here how the transformation has been done with all details.

- Please include a new M&M subsection. “4.8. Statistical analysis”.

- Also, real-time PCR section should be included. Please provide all details on how qRT-PCRs have been conducted and analyzed.

 4) English should be improved (moderate changes needed). Please use a service or help of native speaker to improve the language. Please show all English corrections using MS Word tracking system, so that Reviewers and Editors can see what has been done.

For example, line 195, 214 correct “diferent” to “different”

Also, for example, there are many too complex sententences that need correction to make them more readable. For example, “The VvSnRK2.7 gene was previously screened by transcriptomic data, and subcellular  localization revealed that it was localized in the cell membrane and nucleus, and yeast  two-hybrid screening revealed that the VvbZIP gene interacted with it, and that VvbZIP  might  activate  the  expression  of  VvSnRK2.7.”

5) Figure 8 is mentioned in the manuscript, e.g. line 238-240, but is absent in the manuscript. Where are the results?

Minor edits:

- Abstract. Please provide full SnRK name when SnRK2.7 was firstly mentioned.

- Check all species names to be in italics. For example, Nicotiana tabacum should be in italics in Figure 3 legend or line 334, 341 E. coli.

- Check all gene names to be in italics. For example, line 178, line 216– VvSnRK2.7

- Literature references are given in inappropriate way – ” dipping31”, “transformation32”. Please refer to papers using journal style

Comments on the Quality of English Language

English should be improved (moderate changes needed). Please use a service or help of native speaker to improve the language. Please show all English corrections using MS Word tracking system, so that Reviewers and Editors can see what has been done.

For example, line 195, 214 correct “diferent” to “different”

Also, for example, there are many too complex sententences that need correction to make them more readable. For example, “The VvSnRK2.7 gene was previously screened by transcriptomic data, and subcellular  localization revealed that it was localized in the cell membrane and nucleus, and yeast  two-hybrid screening revealed that the VvbZIP gene interacted with it, and that VvbZIP  might  activate  the  expression  of  VvSnRK2.7.”

Round 2

Reviewer 1 Report

Comments and Suggestions for Authors

The author has addressed all the major concerns, but still needs some minor revisions before its acceptance:

1.    In line 12: Arabidopsis thaliana (Arabidopsis), remove “(Arabidopsis)”

2.    In line 138, Arabidopsis should be italicized.

3.    In line 316, Arabidopsis should be italicized. Authors should be checked throughout the manuscript.

4.    In line 345, A. tumefaciens should be italicized.

5.    In line 355, “The specific primers of VvSnRK2.7 were used to identify transgenic Arabidopsis via PCR”. Refer the primers used for genotyping?

6.    In line 378, Although the author cited Ma et al. [30] and indicated the method used for sugar content analysis, it needs to explain briefly how they prepared the samples and analyzed them for the general readers.

Reviewer 2 Report

Comments and Suggestions for Authors

The manuscript has been considerably improved. The Authors have addressed my comments, except for two issues.

- It is necessary to include information in the M&M on how cDNA library for Y2H has been obtained, including information on how RNA was isolated for cDNA preparation.

- Also, there is no information on how RNA was isolated  for qRT-PCRS.

Please include subsection  "RNA isolation" in the M&M. 

Comments on the Quality of English Language

Minor editing is required.
